SciPost Physics

# An exact method for bosonizing the Fermi surface in arbitrary dimensions

Takamori Park[1][*] and Leon Balents [2, 3]

**1** Department of Physics, University of California, Santa Barbara, CA 93106, USA
**2** Kavli Institute for Theoretical Physics, University of California, Santa Barbara, CA 93106, USA
**3** Canadian Institute for Advanced Research, Toronto, Ontario, Canada
[*] t_park@ucsb.edu

February 1, 2024

## Abstract

Inspired by the recent work by Delacretaz et. al. [1], we rigorously derive an exact and simple method to bosonize a non-interacting fermionic system with a Fermi surface starting from a microscopic Hamiltonian. In the long-wavelength limit, we show that the derived bosonized action is exactly equivalent to the action obtained by Delacretaz et. al. In addition, we propose diagrammatic rules to evaluate correlation functions using our bosonized theory and demonstrate these rules by calculating the three- and four-point density correlation functions. We also consider a general density-density interaction and show that the simplest approximation in our bosonic theory is identical to RPA results.

# 1  Introduction

Quantum phases of matter with an extensive number of gapless modes form an interesting class of phases in condensed matter physics that includes the Fermi liquid phase and the broad class of strongly correlated non-Fermi liquids. While the former is well understood, non-Fermi liquids are a challenge to study because they exist in the strong coupling regime which cannot be accessed using standard perturbative techniques. Instead, non-perturbative approaches are necessary to understand how interactions destroy the quasi-particle picture in non-Fermi liquids. In the past, bosonization of the Fermi surface in $d > 1$ attracted attention because of its potential as a non-perturbative method to study the effects of interactions. This lead to the proposal of several different higher-dimensional bosonization methods [2–16]. However, these approaches involve complicated constructions (e.g. separating the Fermi surface into patches and coarse-graining [3]).

More recently, Delacretaz et. al. proposed a new elegant method to bosonize the Fermi surface that incorporates nonlinear effects to all orders and appears to correctly capture Fermi liquid physics in the long-wavelength limit [1]. Guided by the principle that their action should reproduce the Boltzmann equation, they obtain a bosonic action through a series of inpsired conjectures. Due to the heuristic nature of the derivation some details of their bosonization scheme need clarifying: (a) The bosonized action is obtained in the long-wavelength limit, but how are corrections to this incorporated into the theory? (b) Delacretaz et. al. demonstrated that tree-level diagrams are sufficient to calculate the two- and three-point density correlation functions. Is this generally true for all correlation functions? What do higher-order loop diagrams contribute, and is there a way to connect these bosonic diagrams to their corresponding fermionic diagrams? (c) Delacretaz et. al. incorporate interaction by writing down the most general form it can take. Naively, one would expect a density-density interaction term to contribute the term $\int V f f$ to the action where $V$ is the potential and $f$ is the distribution function, but is this true?

Inspired by the work by Delacretaz et. al. [1], in this work we present an exact method for the bosonization of a system with a Fermi surface. In Section 2, we outline the derivation of our result. Our key insight is to adapt a well known field theory result regarding the effective action that is commonly found in field theory textbooks [17]. Applying this, we are able to exactly express the generating functional as a restricted path integral over a bosonic field. In the long-wavelength limit, the action that we obtained reduces to the action obtained by Delacretaz et. al. We also briefly discuss how interactions modify the functional integral. In Section 3, we propose diagrammatic rules to calculate expectation values diagrammatically and demonstrate their utility by calculating the three- and four-point density correlation functions. Comparison with the fermion loop result is straightforward, and we find our result is exactly correct. We also discuss the effects of interactions and calculate approximate corrections to the two-point density correlation function. Finally, in Section 4, we wrap up our work with a discussion of our results. We discuss the connection between our work and the work by Delacretaz et. al. and provide answers to the questions listed above. In the end we also go over exciting possible future directions of our work.

## 2 Derivation

In this section, we derive a method to bosonize a non-interacting fermion theory with a Fermi surface from an algebraic perspective. We start with a generating functional and first calculate its Legendre transform. Then, we will use a well-known result typically found in field theory textbooks [17] to express the generating functional as a restricted functional integral of its Legendre transform over a bosonic field. At the end of this section, we will also consider how density-density interactions can be incorporated into the bosonized theory.

As an aside, in our derivation we will deal with functions with two spatial and one temporal arguments. Initially, we will represent these functions using the position basis, but this is an arbitrary choice. We could also work in the momentum basis or a mixed basis (the Wigner representation) [18,19] via an appropriate transformation. It is also possible to handle these functions in a basis-free manner by noticing, for example, that a function expressed in the position basis $J(x, x', t)$ can be interpreted as the position-basis matrix element of a first quantized operator, i.e. $\exists J$ such that $\langle x|J(t)|x' \rangle = J(x, x', t)$. As we will see later, this compresses the notation and simplifies switching between different bases.

### 2.1 Generating functional $Z_0[J]$

We start with the generating functional $Z_0[J]$ that generates density matrix correlation functions of a non-interacting fermionic system with a Fermi surface at $T = 0$:

$$Z_0[J] = \langle \Omega | U_{[J]}(\infty, -\infty) | \Omega \rangle. \tag{1}$$

Here, $|\Omega\rangle$ is the many-body ground state representing the Fermi sea and $U_{[J]}$, which is a functional of the source $J$, is the time-evolution operator defined in the usual way as

$$U_{[J]}(t, t') \equiv \begin{cases} \mathbb{T} \exp\left(-i \int_{t'}^{t} d\tau H_0(\tau) - i \int_{t'}^{t} d\tau \int_{x, x'} \psi^\dagger(x) J(x, x', \tau) \psi(x')\right) & t > t' \\ \tilde{\mathbb{T}} \exp\left(-i \int_{t'}^{t} d\tau H_0(\tau) - i \int_{t'}^{t} d\tau \int_{x, x'} \psi^\dagger(x) J(x, x', \tau) \psi(x')\right) & t < t' \end{cases}. \tag{2}$$

$H_0$ is the non-interacting Hamiltonian, and $\mathbb{T}, \tilde{\mathbb{T}}$ denote time ordering and anti-time ordering. We make an important assumption that the source, $J$, satisfies

$$U_{[J]}(\infty, -\infty) | \Omega \rangle \propto | \Omega \rangle. \tag{3}$$

Then, $Z_0[J]$ is simply a complex phase, and if we define the "free energy" $F_0[J]$ such that

$$F_0[J] \equiv i \ln Z_0[J] \quad \left( \implies Z_0[J] = e^{-iF_0[J]} \right), \tag{4}$$

$F_0[J]$ must be real-valued. Using Eq. (3), the functional derivative of $F_0[J]$ is

$$f_{[J]}(x, x', t) \equiv \frac{\delta F_0[J]}{\delta J(x', x, t)} = \langle \Omega | U_{[J]}(-\infty, t) \psi^\dagger(x') \psi(x) U_{[J]}(t, -\infty) | \Omega \rangle \tag{5}$$

which is the one-body reduced density matrix initially in the ground state that evolves under $H_0$ and the source $J$.

Because the Hamiltonian is quadratic in the field operators, the one-body reduced density matrix which we henceforth simply refer to as the density matrix, satisfies the von Neumann equation

$$i \partial_t f_{[J]}(t) = \left[ H_0 + J(t), f_{[J]}(t) \right]. \tag{6}$$

Here and henceforth, $H_0$ refers to the first quantized form of the non-interacting Hamiltonian. In addition, because of the condition on $J$, Eq. (3), the density matrix (expressed in the momentum basis) must satisfy the boundary conditions

$$\lim_{t \to \pm \infty} f_{[J]}(k, k', t) = \langle \Omega | c^\dagger(k') c(k) | \Omega \rangle = f_0(k)(2\pi)^d \delta^d(k - k'), \qquad (7)$$

where $f_0(k) = \theta(k_F - |k|)$ is the zero-temperature Fermi-Dirac distribution function. We assumed a rotationally symmetric Fermi surface for simplicity. $f_0(k)$ only takes the values 0 or 1, so $f_0^2 = f_0$ and $f_0$ must be a projection operator onto states in the Fermi sea. $f_{[J]}$ evolves unitarily from $f_0$, so it must also be a projection operator but onto states in the time-evolved Fermi sea.

## 2.2 Legendre transform of $Z_0[J]$

Now, let us consider how to calculate the Legendre transform of $Z_0[J]$. As we will see, the generating functional does not have a well defined Legendre transform because it is generally not convex [20]. We will resolve this issue by first restricting $Z_0[J]$ to a domain $\mathcal{F}_s$ in which it is convex.

Let us begin by first showing that $Z_0[J]$ is not convex. Consider the map $J \mapsto f_{[J]}$. This map is not injective because given $f_{[J]}$, if we define $\chi_g \equiv f_{[J]} \chi f_{[J]} + (1 - f_{[J]}) \chi (1 - f_{[J]}) \neq 0$ for an aribtrary $\chi$, then $f_{[J]} = f_{[J + \chi_g]}$. The proof is straightforward. First, note that $[\chi_g, f_{[J]}] = 0$ using the property that $f_{[J]}$ is a projection operator. Then, we can add this term to the right-hand side of Eq. (6) to get

$$i \partial_t f_{[J]}(t) = [H_0 + J(t) + \chi_g(t), f_{[J]}(t)], \qquad (8)$$

but this is the equation of motion for $f_{[J + \chi_g]}$. Since, $f_{[J]}$ and $f_{[J + \chi_g]}$ both satisfy the same equation of motion and boundary conditions, it must be true that $f_{[J + \chi_g]} = f_{[J]}$. Therefore, $J \mapsto f_{[J]} = \frac{\delta F[J]}{\delta J}$ is not an injective map, and $F[J]$ is not convex.

The transformation $J \to J + \chi_g$ defines a gauge transformation since $f_{[J]}$, which is the physical degree of freedom, remains invariant under this transformation. If we completely fix this gauge, then $f_{[J]}$ now defines an injective map. Let $\mathcal{F}_s$ denote the space of sources that satisfies this gauge and the condition Eq. (3), and let $\mathcal{F}_d$ denote the set of all density matrices that evolve unitarily in time and satisfy the boundary conditions in Eq. (7). Then, the map $\mathcal{F}_s \ni J \mapsto f_{[J]} \in \mathcal{F}_d$ is bijective, and we can calculate the Legendre transform of $F_0[J]$ defined on $\mathcal{F}_s$.

$f_{[J]}$ defines a bijective map so we can invert it. Given $f \in \mathcal{F}_d$, we define $J_{[f]} \in \mathcal{F}_s$ such that $f_{[J]} \big|_{J = J_{[f]}} = f$. Then, the Legendre transform of $F_0[J]$ is

$$\Gamma_0[f] \equiv \left( F_0[J] - \int_{x,x',t} \frac{\delta F_0[J]}{\delta J(x', x, t)} J(x', x, t) \right) \Bigg|_{J = J_{[f]}} \qquad (9)$$

$$= F_0[J_{[f]}] - \mathrm{Tr}[J_{[f]} f] \qquad (10)$$

where $\Gamma_0[f]$ is defined for $f \in \mathcal{F}_d$. An explicit expression for $J_{[f]}$ is obtained later in Section 2.4. The Legendre transformation is invertible, so there is no loss of information in this transformation. In fact, the Legendre transformation of $\Gamma_0[f]$ is $F_0[J]$.

## 2.3 Expressing $Z_0[J]$ as a functional integral over a bosonic field

In this subsection, we derive an expression for the generating functional as a functional integral over a bosonic field. This is achieved by parametrizing the space of density matrices $\mathcal{F}_d$ using

the Lie algebra of unitary transformations and adapting a field theory result typically found in field theory textbooks concerning the effective action [17].

As we discussed in the previous subsection, the Legendre transformation of $\Gamma_0[f]$ is $F_0[J]$. Using the fact that the leading order term of the saddle-point approximation is equivalent to the Legendre transform, we can write [17, 20]

$$\int_{\mathcal{F}_d} \mathcal{D}f \exp\big(-i\alpha^{-1}(\Gamma_0[f] + \text{Tr}[fJ])\big) \rightarrow \exp\big(-i\alpha^{-1}F_0[J]\big) \qquad \text{as} \qquad \alpha \rightarrow 0. \quad (11)$$

It is easy to verify that the saddle-point value of the exponent on the left-hand side is indeed $F_0[J]$. To evaluate the functional integral in Eq. (11), we need to parametrize $\mathcal{F}_d$. One convenient choice for parametrization is using the Lie algebra of unitary transformations. By definition, we know that $f \in \mathcal{F}_d$ evolves unitarily from $f_0$, so we can always write

$$f(t) = e^{i\phi(t)} f_0 e^{-i\phi(t)} \quad (12)$$

where $e^{i\phi(t)}$ is a unitary transformation and $\phi(t)$ is an element of the Lie algebra of unitary transformations. However, as we discussed in Section 2.2, there is a gauge transformation that introduces redundant descriptions of density matrices. For example, given a unitary transformation $V(t)$ that commutes with $f_0$, it is clear that $e^{i\phi(t)} \rightarrow e^{i\phi(t)}V(t)$ is a gauge transformation. We can fix the gauge by assuming that $\phi \in \mathcal{F}_g$ where $\mathcal{F}_g$ is the space of all unitary Lie algebra elements that satisfy the conditions

$$\forall \phi \in \mathcal{F}_g, f_0 \phi f_0 = (1 - f_0)\phi(1 - f_0) = 0 \quad \text{and} \quad \lim_{t \rightarrow \pm\infty} \phi(t) = 0. \quad (13)$$

The first condition implies $\phi$ only couples momentum states inside and outside of the Fermi sea. There are other ways to fix the gauge, but this is the most convenient. The second condition in Eq. (13) imposes the boundary condition for $f$, Eq. (7).

Using this parametrization, $f = f[\phi]$ is now a functional of $\phi$, and the functional integral in Eq. (11) in the limit of small $\alpha$ is

$$I[J, \alpha] \equiv \int_{\mathcal{F}_g} \mathcal{D}\phi \exp\big(-i\alpha^{-1}(\Gamma_0[f[\phi]] + \text{Tr}[f[\phi]J])\big). \quad (14)$$

The Jacobian $\mathcal{J}[\phi] = \det\left[\frac{\delta f[\phi]}{\delta \phi}\right]$ that comes from the change of variables is a subleading contribution to the action as $\alpha \rightarrow 0$, so it can be ignored. Technically, the unitary group is compact, but extending the integral over $\phi$ to all values only changes the functional integral up to an overall factor. Therefore, $I[J, \alpha]$ can now be treated as a Gaussian integral with perturbations, and it can be evaluated using Feynman diagrams.

Let us consider what type of diagrams we need to consider as $\alpha \rightarrow 0$. It is known that $\alpha$ is a loop expansion parameter: Each vertex contributes a factor of $\alpha^{-1}$ and each propagator contributes a factor of $\alpha$, so Feynman diagrams with $L$ loops are proportional to $\alpha^{L-1}$. Since $I[J, \alpha] \rightarrow \exp\big(-i\alpha^{-1}F_0[J]\big)$ as $\alpha \rightarrow 0$, $-iF_0[J]$ must correspond to connected diagrams with the weight $\alpha^{-1}$ which are tree diagrams (L=0). Therefore,

$$F_0[J] = i \int_{\substack{\mathcal{F}_g \\ \text{connected} \\ \text{tree}}} \mathcal{D}\phi\, e^{-i(\Gamma_0[f[\phi]] - \Gamma_0[f_0]) - i\,\text{Tr}[(f[\phi] - f_0)J]} + \text{Tr}[f_0 J] \quad (15)$$

up to an additive constant, and the generating functional itself is the sum of all tree diagrams (that don't need to be connected),

$$Z_0[J] = \int_{\substack{\mathcal{F}_g \\ \text{tree}}} \mathcal{D}\phi\, e^{-i\Gamma_0[f[\phi]] - i\,\text{Tr}[f[\phi]J]}. \quad (16)$$

We refer to these functional integrals that are limited to certain diagrams as restricted functional integrals. As we have already mentioned, this is an adaptation of a well-known field theory result on effective actions found in field theory textbooks [17].

## 2.4 Bosonized action $S[\phi]$

Here, we define the bosonized action $S[\phi]$ and find an explicit expression for it. We also consider how interactions modify the action.

The definition for the action is given by

$$S[\phi] = -\Gamma_0[f[\phi]] + \Gamma_0[f_0]. \tag{17}$$

In order to find an explicit expression for $\Gamma_0[f[\phi]]$, we first identify the correspondence

$$U_{[J_{[f]}]}(t, -\infty) = e^{i\phi(t)}. \tag{18}$$

There are other ways to relate $U$ and $\phi$ because of the gauge structure, but this is most convenient. Under this correspondence, it is clear that $Z_0[J_{[f]}] = \langle\Omega|U_{[J_{[f]}]}(\infty, -\infty)|\Omega\rangle = 1$ and $F_0[J_{[f]}] = 0$ because of the second condition in Eq. (13). Therefore, the first term in Eq. (10) vanishes. Next, taking the time derivative of Eq. (18) gives us $i\partial_t e^{i\phi(t)} = (H_0 + J_{f[t]}(t))e^{i\phi(t)}$, so

$$J_{[f]} = i(\partial_t e^{i\phi(t)})e^{-i\phi(t)} - H_0. \tag{19}$$

Therefore, the action must be

$$S[\phi] = \text{Tr}\big[J_{[f]}f[\phi]\big] + \text{Tr}[H_0 f_0] = \text{Tr}\big[f_0 e^{-i\phi}(i\partial_t - H_0)e^{i\phi}\big] + \text{Tr}[f_0 H_0]. \tag{20}$$

The free energy and generating functional is given by

$$F_0[J] = i \int_{\substack{\mathcal{F}_g \\ \text{connected tree}}} \mathcal{D}\phi\, e^{iS[\phi] - i\,\text{Tr}[(f[\phi] - f_0)J]} + \text{Tr}[f_0 J] \tag{21}$$

$$Z_0[J] = \int_{\substack{\mathcal{F}_g \\ \text{tree}}} \mathcal{D}\phi\, e^{iS[\phi] - i\,\text{Tr}[f[\phi]J]}. \tag{22}$$

Up until now, we have exclusively considered non-interacting systems, but incorporating interactions is straightforward. Consider a general two-body interaction

$$H_{\text{int}} = \frac{1}{2}\int_{x,y} V(x-y)\psi^\dagger(x)\psi^\dagger(y)\psi(y)\psi(x). \tag{23}$$

The generating functional of the interacting system $Z[J]$ can be expressed using its non-interacting counter part according to

$$Z[J] = \exp\left(\frac{i}{2}\int_{x,y,t} V(x-y)\frac{\delta}{\delta J(x,x,t)}\frac{\delta}{\delta J(y,y,t)}\right)Z_0[J]. \tag{24}$$

This adds an additional term to the action $S[\phi] \to S[\phi] + S_{\text{int}}[\phi]$ where

$$S_{\text{int}}[\phi] = -\frac{1}{2}\int_{x,y,t} V(x-y)f[\phi](x,x,t)f[\phi](y,y,t) \tag{25}$$

$$= -\frac{1}{2}\int_{q,k,k',t} \tilde{V}_q f[\phi](k+q,k,t)f[\phi](k'-q,k',t). \tag{26}$$

The action that we obtained (Eq. (20)) looks similar to the action derived by Delacretaz et. al. [1]. In Section 4 and Appendix A we discuss the link between the two actions in detail.

# 3 Diagrammatic calculation of correlation functions

In this section, we propose a set of diagrammatic rules that simplifies the task of calculating correlation functions from the bosonic theory. We demonstrate their use by calculating the three- and four-point density correlation functions. Since the bosonic theory is exact, we can directly compare results with the equivalent fermionic theory. We find that the diagrams obtained from the bosonic theory are partitions of the corresponding diagrams obtained from the fermionic theory. In other words, the process of bosonization separates each fermion diagrams into several bosonic diagrams. In the long wavelength limit, fermion 1-loop diagrams exhibit partial cancellation and extracting the correct leading order behavior can be tricky [21, 22]. However, we find that the diagrams calculated from the bosonic theory are well-behaved, and calculating the correct long wavelength behavior becomes straightforward.

## 3.1 Expanding the action and density matrix in powers of $\phi$

We first begin by expanding the action given in Eq. (20) in powers of $\phi$. We find

$$S[\phi] = \sum_{n=1}^{\infty} \frac{(-i)^n}{n!} \text{Tr}\Big[ f_0\Big( -i \text{ad}_\phi^{n-1} \partial_t \phi - \text{ad}_\phi^n H_0 \Big) \Big], \tag{27}$$

where $\text{ad}_\phi(\cdot) \equiv [\phi, \cdot]$ is the adjoint action of the Lie algebra. The action is most conveniently expressed in the momentum basis since $H_0$ and $f_0$ are diagonal in this basis. The symmetrized action in the momentum basis is

$$S[\phi] = \sum_{\substack{n=2 \\ n:\text{even}}}^{\infty} S_{(n)}[\phi], \tag{28}$$

where

$$S_{(n)}[\phi] \equiv \frac{i^n 2^{n-2}}{n!} \int_{\substack{p_1,\cdots,p_n \\ \omega_1,\cdots,\omega_n}} \left[ \frac{-\omega_1 + \omega_2 - \cdots + \omega_n}{n} - 2 \frac{-\varepsilon_{p_1} + \varepsilon_{p_2} - \cdots + \varepsilon_{p_n}}{n} \right] F^{(n)}(p_1, \cdots, p_n)$$

$$\times \phi(p_1, p_2, \omega_1) \cdots \phi(p_n, p_1, \omega_n) 2\pi \delta(\omega_1 + \cdots + \omega_n). \tag{29}$$

$\varepsilon_p = \langle p | H_0 | p \rangle$ is the dispersion which for simplicity we have assumed is $\varepsilon_p = \frac{p^2}{2m} - E_F$. The factor $F^{(n)}$ is a function that only takes the values $-1, 0, 1$ and is defined as

$$F^{(n)}(p_1, \cdots, p_n) \equiv \begin{cases} f_0(p_1)[1 - f_0(p_2)]f_0(p_3) \cdots f_0(p_n) \\ \quad - [1 - f_0(p_0)]f_0(p_1)[1 - f_0(p_2)] \cdots [1 - f_0(p_n)] & n: \text{even} \\ [1 - f_0(p_1)]f_0(p_2)[1 - f_0(p_3)] \cdots f_0(p_n) \\ \quad - f_0(p_1)[1 - f_0(p_2)]f_0(p_3) \cdots [1 - f_0(p_n)] & n: \text{odd} \end{cases}. \tag{30}$$

Adjacent momenta, $p_i, p_{i+1}$, in the argument cannot both be in or out of the Fermi sea. This is a consequence of the first condition in Eq. (13). For example, if $f_0(p_i) = f_0(p_{i+1})$, then $F^{(n)}(p_1, \cdots, p_n) = 0$. In addition, as a result of this condition, odd $n$ terms in the action must vanish because they are proportional to $f_0(p_1)(1 - f_0(p_1))$ which vanishes because $f_0$ is a projection operator.

The quadratic contribution to the action is

$$S_0[\phi] \equiv S_{(2)}[\phi] = \int_{k,k'} \frac{1}{2} \big[ \omega - \varepsilon_k + \varepsilon'_k \big] \big| \phi(k, k', \omega) \big|^2 F^{(2)}(k, k'). \tag{31}$$

This allows us to define the propagator evaluated with respect to $S_0$

$$\langle \phi(k,k',\omega)\phi(p',p,\omega')\rangle_0 = D_0(k,k',\omega)(2\pi)^{2d+1}\delta^d(p-k)\delta^d(p'-k')\delta(\omega+\omega'). \tag{32}$$

where

$$D_0(k,k',\omega) = D_0(k',k,-\omega) \equiv -i\frac{f_0(k)-f_0(k')}{\omega-\varepsilon_k+\varepsilon_{k'}} \tag{33}$$

Here, we used the fact that, $F^{(2)}(k,k') = f_0(k')-f_0(k)$. Note, $\langle\cdots\rangle_0$ indicates evaluation with respect to $S_0$.

Next, starting from our choice of the definition $f(t) = e^{i\phi(t)}f_0 e^{-i\phi(t)}$, we also expand the density matrix in powers of $\phi$ and find

$$f[\phi](t) = \sum_{n=0}^{\infty}\frac{i^n}{n!}\mathrm{ad}_{\phi(t)}^n f_0. \tag{34}$$

In the momentum basis, the density matrix is expressed as

$$f[\phi](k,k',t) = \sum_{n=0}^{\infty}f_{(n)}[\phi](k,k',t) \tag{35}$$

where

$$f_{(n)}(k,k',t) \equiv \frac{i^n 2^{n-1}}{n!}\int_{k_1,\cdots,k_{n-1}}\phi(k,k_1,t)\phi(k_1,k_2,t)\cdots\phi(k_{n-1},k',t)F^{(n+1)}(k,k_1,\cdots,k_{n-1},k'). \tag{36}$$

Lastly, let us go over how to evaluate correlation functions. Consider a general correlation function of the form

$$\langle f(k_1,k_1',\omega_1)\cdots f(k_n,k_n',\omega_n)\rangle. \tag{37}$$

The general procedure to evaluate such a correlation function is to expand the action and each $f$ in powers of $\phi$ and keep only the tree diagrams. Explicitly,

$$\langle f(k_1,k_1',\omega_1)\cdots f(k_N,k_N',\omega_N)\rangle$$
$$= \sum_{n_1=0}^{\infty}\cdots\sum_{n_N=0}^{\infty}\sum_{\substack{m=4\\m:\text{even}}}^{\infty}\langle f_{(n_1)}(k_1,k_1',\omega_1)\cdots f_{(n_N)}(k_N,k_N',\omega_N)iS_{(m)}\rangle_{0,c,\text{tree}} \tag{38}$$

Even though, the expansion contains an infinite number of terms, since the expectation value is restricted to tree diagrams, in reality we only need to consider a finite number of terms.

## 3.2 Diagrammatic rules

Correlation functions can be calculated using the bosonized theory without the use of diagrams, but they often involve lengthy and tedious algebra resulting from integrating over multiple delta functions. This is because the field $\phi$ has two spatial indices rather than one which doubles the number of spatial delta functions. This can be avoided by using Feynman diagrams. Below, we establish Feynman diagram rules for the bosonized theory:

- Points represent momenta.

- The field $\phi(k,k',\omega)$ is represented by a line with an arrow that starts at $k'$ and ends at $k$. A dashed line with an arrow is used to keep track of the frequency and momentum exchange. (Fig. 1a)

- The propagator $D_0(k,k',\omega)$ is represented by a loop connecting the momenta $k$ and $k'$. (Fig. 1b)

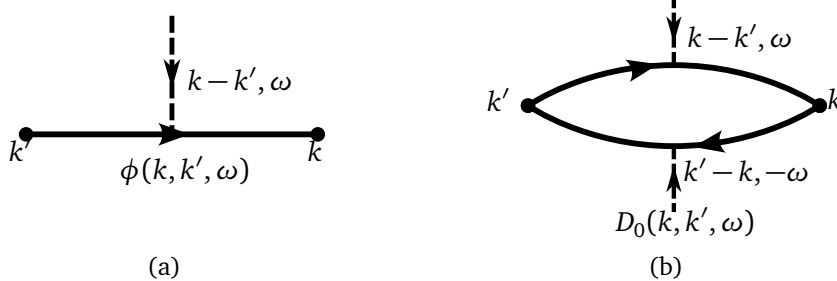

Figure 1: Feynman diagram elements for (a) the field $\phi(k, k', \omega)$ and (b) the propagator $D_0(k, k', \omega)$.

## 3.3 Three-point density correlation function

In this section, we calculate the three-point density correlation function using diagrams of the bosonized theory. For simplicity, we ignore the effects of interaction. The correlation function is defined as

$$C_{(3)}(1, 2, 3) = \langle \rho(1)\rho(2)\rho(3)\rangle_c, \tag{39}$$

where we used the shorthand $i = (q_i, \omega_i)$ and $\rho(i) = \int_k f(k + q_i, k, \omega_i)$ for $i = 1, 2, 3$. We define the $n$-th order term of the expansion of $\rho$ in powers of $\phi$ as $\rho_{(n)}(i) \equiv \int_k f_{(n)}(k + q_i, k, \omega_i)$. In order to expand As we showed above, only the connected tree diagrams need to be considered. Schematically, there is only one such term as shown in Fig. 2a, so

$$C_{(3)}(1, 2, 3) = \left\langle \rho_{(2)}(1)\rho_{(1)}(2)\rho_{(1)}(3)\right\rangle_{0,c} + \left\langle \rho_{(1)}(1)\rho_{(2)}(2)\rho_{(1)}(3)\right\rangle_{0,c} + \left\langle \rho_{(1)}(1)\rho_{(1)}(2)\rho_{(2)}(3)\right\rangle_{0,c}. \tag{40}$$

We evaluate the first term diagramatically. The other two terms are obtained by permuting the indices of the arguments. The first term generates two diagrams that are related by exchange of $(q_2, \omega_2) \longleftrightarrow (q_3, \omega_3)$ as shown in Fig. 3. The diagrams evaluate to

$$\left\langle \rho_{(2)}(1)\rho_{(1)}(2)\rho_{(1)}(3)\right\rangle_{0,c} = -\int_k \left( \frac{F^{(3)}(k + q_1, k + q_1 + q_2, k)}{(\omega_2 - \varepsilon_{k+q_1+q_2} + \varepsilon_{k+q_1})(\omega_3 - \varepsilon_k + \varepsilon_{k+q_1+q_2})} \right.$$
$$+ \frac{F^{(3)}(k + q_1, k + q_1 + q_3, k)}{(\omega_2 - \varepsilon_k + \varepsilon_{k+q_1+q_3})(\omega_3 - \varepsilon_{k+q_1+q_3} + \varepsilon_{k+q_1})} \right)$$
$$\times (2\pi)^{d+1}\delta^d(q_1 + q_2 + q_3)\delta(\omega_1 + \omega_2 + \omega_3). \tag{41}$$

The first and second terms correspond to Fig. 3a and Fig. 3b respectively. By permuting the external indices and reorganizing the terms, we find

$$C_{(3)}(1, 2, 3) = -\int_k \left( \frac{F^{(3)}(k + q_1, k + q_1 + q_2, k)}{(\omega_3 - \varepsilon_k + \varepsilon_{k+q_1+q_2})(\omega_2 - \varepsilon_{k+q_1+q_2} + \varepsilon_{k+q_1})} \right.$$
$$+ \frac{F^{(3)}(k + q_2, k + q_2 + q_3, k)}{(\omega_1 - \varepsilon_k + \varepsilon_{k+q_2+q_3})(\omega_3 - \varepsilon_{k+q_2+q_3} + \varepsilon_{k+q_2})}$$
$$+ \frac{F^{(3)}(k + q_3, k + q_1 + q_3, k)}{(\omega_2 - \varepsilon_k + \varepsilon_{k+q_1+q_3})(\omega_1 - \varepsilon_{k+q_1+q_3} + \varepsilon_{k+q_3})} + (2 \leftrightarrow 3) \right). \tag{42}$$

Here, we omitted the delta function factors that conserve momentum and frequency. The first three terms correspond to the fermion loop diagram that describe the scattering process, $k \to k + q_1 \to k + q_1 + q_2 \to k$ which we refer to as the (123) scattering process. The $(2 \leftrightarrow 3)$ term corresponds to the (132) scattering process which is the reversed process. In Appendix B, we show that Eq. (42) exactly matches the corresponding fermion loop calculation.

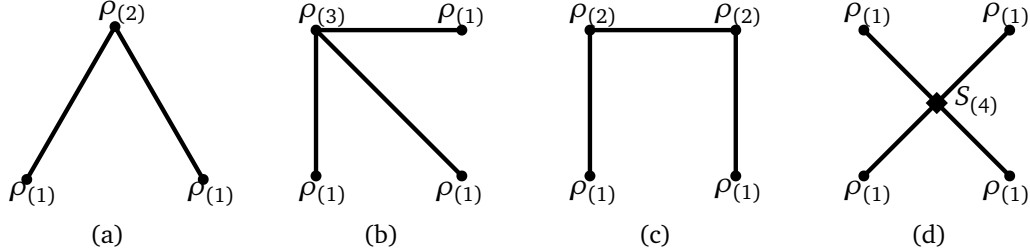

Figure 2: Schematic tree-level diagrams for the three- and four-point density correlation functions. (a) corresponds to the three-point correlation function and (b–e) correspond to the four-point correlation function.

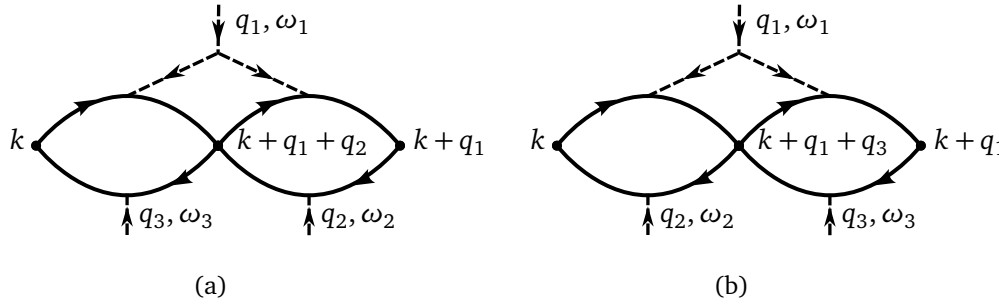

(a)            (b)

Figure 3: Feynman diagrams for $\left\langle \rho_{(2)}(1)\rho_{(1)}(2)\rho_{(1)}(3) \right\rangle_{0,c}$.

Notice, that in the low-momentum limit, $F^{(3)}$ in the numerator of each term vanishes if $k$ is not near the Fermi surface. Moreover, if we take the limit $|q_i| \to 0$, the $F^{(3)}$ factors becomes a delta function $\delta(|k| - k_F)$ and the integral over $k$ is reduced to an angular integral. The leading and sub-leading behavior can be straightforwardly extracted from this.

Let us study the long-wavelength behavior of the correlation function. As we mentioned above, in the original fermionic theory, partial cancellation between fermion loop diagrams occur when $\omega_i \to 0, q_i \to 0$, and the long-wavelength behavior of the density correlation function is determined by the sub-leading term [21,22]. This cancellation can be tricky to see in the original theory, but we demonstrate it is straightforward to see in our bosonic theory. Consider the term Eq. (41). The numerator of the first term expands to

$$F^{(3)}(k+q_1, k+q_1+q_2, k) = f_0(k) - f_0(k+q_2) + \left( \begin{smallmatrix} \text{terms symmetric under} \\ \text{permutation of} \end{smallmatrix} \{q_1, q_1+q_2, q_1+q_2+q_3\} \right). \tag{43}$$

The symmetric term vanishes if we consider all other diagrams, so we only need to consider the first two terms. (This is also explained in Appendix B.) The leading order behavior of the first term in Eq. (41) is

$$\int_k \frac{q_2 \cdot \hat{k}\delta(|k| - k_F)}{(\omega_2 - q_2 \cdot \nabla_k \varepsilon_k)(\omega_3 + (q_1 + q_2) \cdot \nabla_k \varepsilon_k)}. \tag{44}$$

Clearly this diverges as $q_i, \omega_i \to 0$, but notice that the first and second terms in Eq. (41) are related by a change in sign of the external momenta and frequency, $q_i, \omega_i \to -q_i, -\omega_i$. This tells us that the leading order behavior of the second term as $q_i, \omega_i \to 0$ is given by the negative of Eq. (44). Therefore, the leading order behavior of the two terms in Eq. (41) cancel and their sum does not diverge in the long-wavelength limit. Instead, the sub-leading terms of the two terms in Eq. (41) determines the scaling.

### 3.4 Four-point density correlation function

Let us now consider the four-point density correlation function defined as

$$C_{(4)}(1,2,3,4) = \langle \rho(1)\rho(2)\rho(3)\rho(4)\rangle_c. \tag{45}$$

The three possible connected tree diagrams are schematically shown in Figs. 2b–2d. They correspond to the terms

$$\begin{aligned}
C_{(4)}(1,2,3,4) =& \Big(\langle \rho_{(3)}(1)\rho_{(1)}(2)\rho_{(1)}(3)\rho_{(1)}(4)\rangle_{0,c} + \text{(other permutations)}\Big) \\
&+ \Big(\langle \rho_{(2)}(1)\rho_{(2)}(2)\rho_{(1)}(3)\rho_{(1)}(4)\rangle_{0,c} + \text{(other permutations)}\Big) \\
&+ \langle \rho_{(1)}(1)\rho_{(1)}(2)\rho_{(1)}(3)\rho_{(1)}(4)iS_{(4)}\rangle_{0,c}.
\end{aligned} \tag{46}$$

To simplify the comparison with the corresponding fermion loop diagram, we organize the diagrams by the scattering process they describe. There are $3! = 6$ distinct scattering processes. For simplicity, we only consider diagrams that describe the fermion scattering process, $k \to k+q_1 \to k+q_1+q_2 \to k+q_1+q_2+q_3 \to k$ which we refer to as the (1234) scattering process as we did in the previous subsection. The other diagrams can be obtained by permuting the external momenta and frequencies. It's straightforward to iterate through the diagrams that contribute to the (1234) scattering process.

The first term in Eq. (46) gives us the diagram Fig. 4a. The second term leads to three diagrams Figs. 4b–4c, and the last term gives us Fig. 4d.

Let

$$g_{ij}(k) = g_{ji}(k) \equiv \frac{f_0(k+p_i) - f_0(k+p_j)}{\Omega_i - \Omega_j - \varepsilon(k+p_i) + \varepsilon(k+p_j)}, \tag{47}$$

where $p_i \equiv \sum_{j=1}^{i} q_j$ and $\Omega_i \equiv \sum_{j=1}^{i} \omega_i$ for $i = 1, \cdots, 4$. The indices are defined mod 4 for convenience. All the diagrams in Fig. 4 can be expressed as an integral over products of these functions. Below, we list the contribution from each diagram.

$$(\text{Fig. 4a}) = -i\frac{2}{3}\sum_{i=1}^{4}\int_k g_{i,i+1}(k)g_{i+1,i+2}(k)g_{i+2,i+3}(k) \tag{48}$$

$$(\text{Fig. 4b}) = -i\sum_{i=1}^{2}\int_k g_{i+3,i+2}(k)g_{i+2,i}(k)g_{i,i+1}(k) \tag{49}$$

$$(\text{Fig. 4c}) = -i\sum_{i=1}^{2}\int_k g_{i,i+1}(k)g_{i+1,i+3}(k)g_{i+3,i+2}(k) \tag{50}$$

$$(\text{Fig. 4d}) = i\sum_{i=1}^{4}\int_k g_{i+1,i}(k)g_{i+2,i}(k)g_{i+3,i}(k) \tag{51}$$

$$(\text{Fig. 4e}) = \frac{i}{6}\sum_{i=1}^{4}\int_k g_{i,i+1}(k)g_{i+1,i+2}(k)g_{i+2,i+3}(k) \tag{52}$$

The factor $(2\pi)^{d+1}\delta^d(q_1+q_2+q_3+q_4)\delta(\omega_1+\omega_2+\omega_3+\omega_4)$ was omitted. Notice, that Eq. (48) and Eq. (52) are identical up to an overall factor. Adding all these contributions together gives us

$$\begin{aligned}
C_{(4)}(1,2,3,4) =& \Big(-\frac{i}{2}\sum_{i=1}^{4}\int_k g_{i+3,i+2}(k)g_{i+2,i+1}(k)g_{i+1,i}(k) + i\sum_{i=1}^{4}\int_k g_{i+1,i}(k)g_{i+2,i}(k)g_{i+3,i}(k) \\
&- i\sum_{i=1}^{2}\int_k g_{i+3,i+2}(k)g_{i+2,i}(k)g_{i,i+1}(k) - i\sum_{i=1}^{2}\int_k g_{i+2,i+3}(k)g_{i+3,i+1}(k)g_{i+1,i}(k)\Big) \\
&+ (\text{permutations of } \{(\omega_1,q_1),(\omega_2,q_2),(\omega_3,q_3),(\omega_4,q_4)\})
\end{aligned} \tag{53}$$

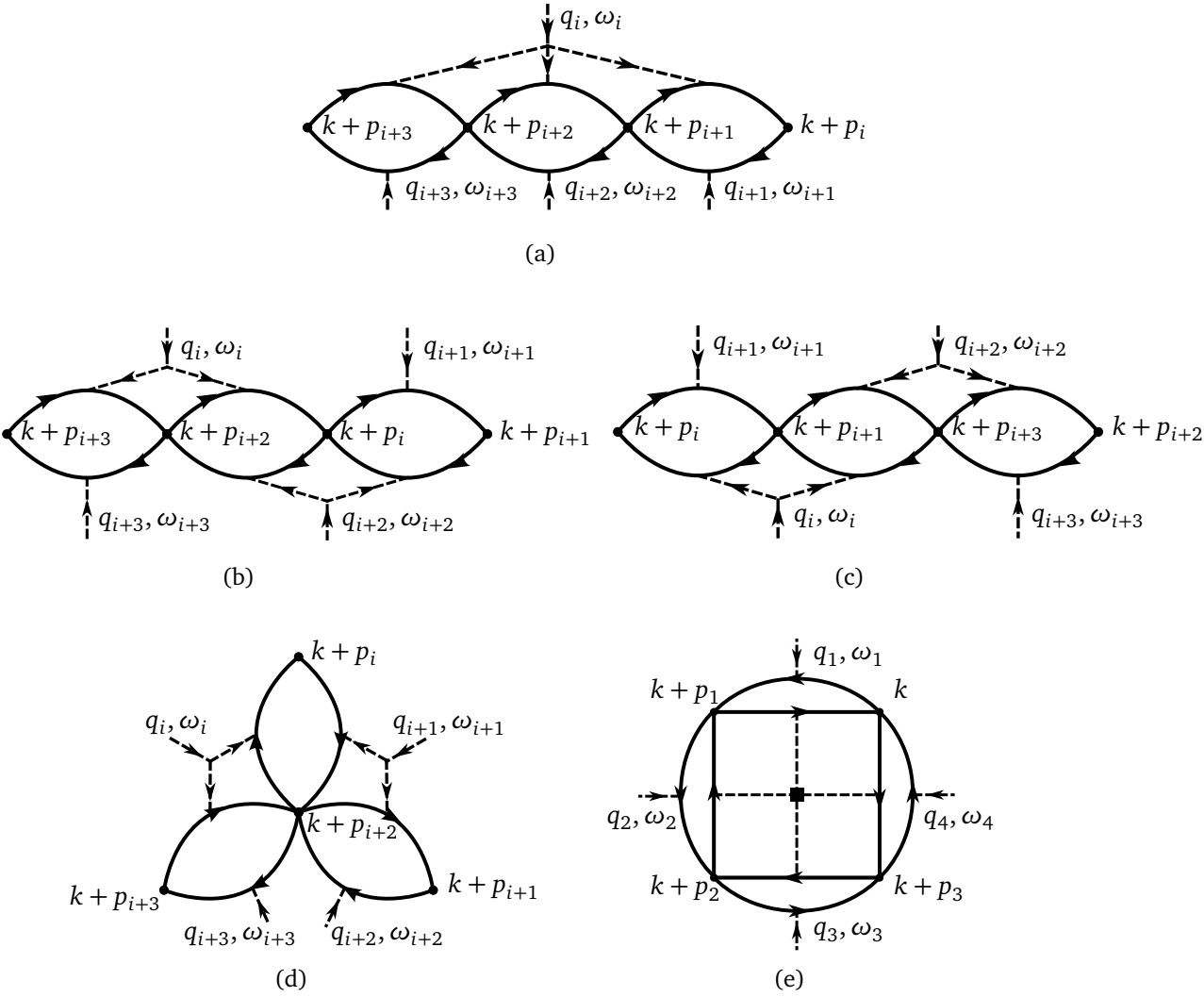

Figure 4: Feynman diagrams of $C_{(4)}$ describing the (1234) scattering process. (a) corresponds to terms of the form $\langle \rho_{(3)}\rho_{(1)}\rho_{(1)}\rho_{(1)}\rangle_{0,c}$, (b–d) descend from terms of the form $\langle \rho_{(2)}\rho_{(2)}\rho_{(1)}\rho_{(1)}\rangle_{0,c}$, and (e) corresponds to $\langle \rho_{(1)}\rho_{(1)}\rho_{(1)}\rho_{(1)}iS_{(4)}\rangle_{0,c}$

where we omitted the overall delta function factors that conserve momentum and frequency. The first four terms correspond to the (1234) scattering process. In Appendix B, we show that Eq. (53) equals the corresponding fermion loop calculation exactly.

## 3.5  Self-energy calculations

Up till now we have ignored interaction, so let us now consider its effects. As we discussed above, a general two-particle interaction introduces the term Eq. (26) to the action. Expanding in powers of $\phi$, we define

$$S_{\text{int}}^{(n,n')} = S_{\text{int}}^{(n',n)} = -\frac{1}{2}\int_{q,k,k',t}\tilde{V}_q f_{(n)}(k+q,k,t)f_{(n')}(k'-q,k',t) \tag{54}$$

so that $S_{\text{int}} = \sum_{n,n'=1}^{\infty} S_{\text{int}}^{(n,n')}$. Let us now think about what kind of diagrams can be drawn if we include these interaction terms. The interaction term is introduced by taking a second-order functional derivative with respect to $J$, so diagrammatically, interactions can be thought of

as a diagram element that stitches together two external legs of a tree diagram. Here, when we refer to external legs we refer to the dashed lines in the full diagrams or the points in the schematic diagrams. The external legs can be from the same tree diagram or from different tree diagrams.

For simplicity, let us consider the two-point density correlation function. The corrections to the correlation function up to second order in interaction are shown in Fig. 5. Each circle, depending on the number of legs it possesses, represents a two-, three-, or four-point correlation function of the *bare* theory. It is clear, that higher-order corrections will involve higher-order correlation functions of the bare fermion.

Let us consider a simple approximation where we only consider diagrams that involve two-point correlation functions. It is then straightforward to show that the two-point density correlation function of the interacting theory under this approximation is

$$C_{(2),\text{int}}(1,2) \approx \int_k D_0(k+q_1,k,\omega_1) \sum_{n=0}^{\infty} \left(-i\tilde{V}_q \int_k D_0(k+q_1,k,\omega_1)\right)^n \tag{55}$$

$$= \frac{\int_k D_0(k+q_1,k,\omega_1)}{1+i\tilde{V}_q \int_k D_0(k+q_1,k,\omega_1)} \tag{56}$$

where the momentum and frequency delta functions are omitted. This is equivalent to the RPA approximation.

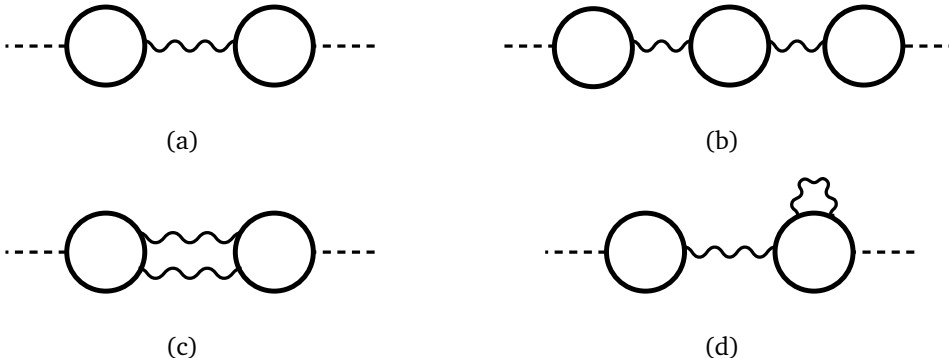

(a)

(b)

(c)

(d)

Figure 5: Schematic representations of the corrections to the two-point density correlation function up to second order in interaction. Here, we used a different diagrammatic convention from before. Circles represent the bare two-, three-, and four-point density correlation functions depending on the number of legs they have. The wavy lines represent the stitching of two legs by interaction.

## 4 Discussion

In this work, we derived an exact method to express a fermionic theory with a Fermi surface as a restricted functional integral over a bosonic field. Starting from a generating functional with a source coupled to density, we calculated the Legendre transform of the free energy. We then showed that the integral of the Legendre-transformed free energy over the Lie algebra of the unitary group when restricted to tree level diagrams reproduces the generating functional. In order to demonstrate the utility of our formalism, we evaluated the three- and four-point density correlation functions using Feynman diagram rules that we proposed. The results from our bosonized theory exactly matches results obtained from fermion loop calculations.

This work was inspired by the work by Delacretaz et. al. [1]. Using heuristic arguments, they derived an elegant bosonization scheme for a system with a Fermi surface that incorporates nonlinear effects. It turns out that the long-wavelength limit of the bosonized action that we derived, Eq. (20), is equivalent to the bosonic action in Ref. [1]. We can see this correspondence by expressing our bosonized action in the Wigner representation. In this representation, operator products are replaced by the non-commutative Moyal product. The action derived by Delacretaz et. al. can then be obtained by expanding the Moyal product to first order in spatial derivatives. Then, the Lie algebra of the unitary group becomes the Poisson algebra of the canonical transformations and the two actions become equivalent. Therefore, our work serves as a rigorous derivation of the results presented in Ref. [1]. In Appendix A we go over this argument in more detail.

In addition to providing a proof, our work also clarifies ambiguities in Ref. [1] that we listed earlier in Section 1. First of all, the bosonized theory in Ref. [1] is valid in the long-wavelength limit, so a question one can ask is how can we obtain corrections to this limit? As discussed above, the long-wavelength limit is a result of using the Poisson algebra so corrections can be obtained by expanding to the Lie algebra of the unitary group. Our bosonized action is exact and considers fluctuations of the Fermi surface generated by the unitary group, so our action can be be used to obtain these corrections. Secondly, Ref. [1] does not explain why correlation functions can be calculated from only tree diagrams. We explained in Section 2 that this is due to taking the saddle-point approximation to obtain the Legendre transform. Lastly, Ref. [1] did not explicitly derive the link between a general four-fermion interaction term and the interaction term in their bosonized action. We derived this in Section 2.

The bosonization formalism in this paper adapts an old field theory result concerning the effective action typically found in field theory textbooks [17]. Naturally, one might ask what is new in this work. This result on the effective action usually applies to bosonic systems with a source coupled linearly to the field, and the Legendre transformation of this action produces another bosonic action. However, in this work we started with a fermionic theory and coupled the source to a fermion bilinear so that the Legendre transform we calculated produces a bosonic action from a fermionic action. In addition, the change of variables from the density matrices to the Lie algebra of the unitary group is a non-linear transformation that partitions the original fermion diagrams in a non-trivial manner. These aspects of our formalism are well beyond the scope of the aforementioned textbook result.

In many ways the bosonization formulation derived in this paper appears similar to standard one-dimensional (1D) bosonization. For one thing, if we take the second-order contribution to the action in 1D and express it in the Wigner representation, to lowest order of the gradient expansion we get $S_{(2)} \approx \frac{1}{2} \sum_{s=\pm} \int_{t,x} (\partial_x \phi_s \partial_t \phi_s + v_F (\partial_x \phi_s)^2)$ where $\phi_\pm(x,t) \equiv \phi(x, \pm k_F, t)$ [1]. This is the bosonized action of non-interacting 1D fermions with linear dispersion [23]. In addition, the calculation of an equal-time Green's function in our formalism, schematically, is $G \sim \langle \psi^\dagger \psi \rangle \sim \langle f \rangle \sim \langle e^{i\phi} f_0 e^{-i\phi} \rangle$ for any dimension. The last expression is reminiscent of a Green's function calculation in standard 1D bosonization where $e^{\pm i\phi}$ plays the role of a vertex operator. However, despite their similarities, the two bosonization formalisms are distinct. Under standard 1D bosonization, the action of 1D fermions with an exactly linear dispersion is strictly quadratic, but under our bosonization formalism, it will generally have an infinite number of higher-order terms. In addition, our formalism has an artificially introduced parameter $\alpha$ that is absent in standard 1D bosonization. This means that in our formalism $G \sim \langle e^{i\phi} f_0 e^{-i\phi} \rangle$ cannot be naively evaluated as an exponential of the propagator of $\phi$ and the anomalous dimension cannot be extracted this way as is done in 1D bosonization. Therefore, it is clear that the two approaches are distinct despite some similarities, and further investigation is required to better understand their connection.

In our work, we briefly discussed the effects of interaction, and we obtained the RPA ap-

proximation by only considering the simplest series of diagrams that contribute to the two-point density correlation function. In general, because the bosonization scheme partitions fermion diagrams into several bosonic diagrams, this increases the variety of diagrams that we need to consider in the bosonized theory. In future works, it would be of value to consider all of these diagrams and see how they can be organized differently compared to the original fermionic theory.

Another direction to consider in the future is to apply the bosonization method we outlined in this work to other systems with different relevant degrees of freedom. In principle, our approach can be simply generalized to different cases by coupling the source $J$ in the generating functional to different fermion bilinears. For example, $J$ can couple to $\psi(x,t)\psi(x',t)$ to describe a BCS theory or to $\psi_\alpha^\dagger(x,t)\frac{\vec{\sigma}_{\alpha\beta}}{2}\psi_\beta(x',t)$ where $\alpha, \beta = \uparrow, \downarrow$ to describe spin fluctuations. One can also consider the case where the source is coupled to two fermion operators at different times, $\psi^\dagger(x,t)\psi(x',t')$. This was considered by Han et. al. in their proposal of a bosonization procedure for non-Fermi liquids in Ref. [24].

## Acknowledgements

The authors thank Yi-Hsien Du, Léo Mangeolle, and Lucile Savary for insightful discussions. T.P. was supported by a quantum Foundry fellowship through the National Science Foundation through Enabling Quantum Leap: Convergent Accelerated Discovery Foundries for Quantum Materials Science, Engineering and Information (Q-AMASE-i) award number DMR-1906325, supplemented by the NSF CMMT program under Grant No. DMR-2116515, which also supported L.B.

## A  Obtaining the action derived by Delacretaz et. al. [1] from Eq. (20)

Here, we show how to explicitly obtain the action derived by Delacretez et. al.in Ref. [1]. For convenience we replicate the action in Eq. (20) here.

$$S[\phi] \equiv -\Gamma[f[\phi]] + \Gamma[f_0] = \text{Tr}\big[f_0 e^{-i\phi}(i\partial_t - H_0)e^{i\phi}\big] + \text{Tr}[f_0 H_0]. \tag{57}$$

To obtain the action in Ref. [1], we need to express the equation above in the Wigner representation. The Wigner representation is a mixed basis representation. Given a single-particle operator $\hat{A}$, its Wigner representation is defined as [18, 19]

$$\hat{A} \to A(x,p) = \int_y e^{-iy\cdot p}\left\langle x + \frac{y}{2}\Big|\hat{A}\Big|x - \frac{y}{2}\right\rangle. \tag{58}$$

Here, a hat is added to operators to emphasize their distinction from c-numbers, but in the main text we do not. The product of operators can be represented as the Moyal product ($\star$-product) of their Wigner representations.

$$\hat{A}\hat{B} \to A(x,p) \star B(x,p) \equiv A(x,p)e^{\frac{i}{2}\left(\overleftarrow{\nabla}_x\cdot\overrightarrow{\nabla}_p - \overleftarrow{\nabla}_p\cdot\overrightarrow{\nabla}_x\right)}B(x,p) \tag{59}$$

The arrows of the derivatives indicate the directions in which they act.

The trace of an operator is the integral of its Wigner representation over phase space.

$$\text{tr}\big[\hat{A}\big] = \int_{x,p} A(x,p) \tag{60}$$

The trace of the product of two operators is the phase-space integral of the products of their Wigner representation.

$$\text{tr}\big[\hat{A}\hat{B}\big] = \int_{x,p} A(x,p)B(x,p) \tag{61}$$

The commutator of two operators in the Wigner representation becomes the Moyal bracket.

$$\big[\hat{A},\hat{B}\big] \rightarrow \{\{A(x,p),B(x,p)\}\} = A(x,p)\star B(x,p) - B(x,p)\star A(x,p). \tag{62}$$

The Moyal bracket defines a Lie algebra which we refer to as the $\star$-algebra. If $A(x,p)$ and $B(x,p)$ vary slowly in space, we can expand the Moyal product in powers of the gradient. The lowest order term of the commutator in the gradient expansion turns out to be the Poisson bracket.

$$\{\{A(x,p),B(x,p)\}\} \approx A(x,p)i\Big(\overleftarrow{\nabla}_x \cdot \overrightarrow{\nabla}_p - \overleftarrow{\nabla}_p \cdot \overrightarrow{\nabla}_x\Big)B(x,p) = i\{A(x,p),B(x,p)\}_{\text{p.b.}} \tag{63}$$

Hence, the Poisson algebra is obtained from the $\star$-algebra in the long-wavelength limit.

Now that we have reviewed the necessary facts on the Wigner representation needed to proceed, let us consider the Hamiltonian term of the action. As we discussed in Section 3, it can be expanded using the adjoint action of $\phi$.

$$e^{-i\phi}H_0 e^{i\phi} = \sum_{n=0} \frac{(-i)^n}{n!} \text{ad}_\phi^n H_0. \tag{64}$$

In the Wigner representation this becomes

$$(e^{-i\phi}H_0 e^{i\phi})(x,p) = \sum_{n=0}^{\infty} \frac{(-i)^n}{n!}\Big(\text{ad}_{\phi(x,p)}^{(\star)}\Big)^n H_0(p) \tag{65}$$

where $\text{ad}_{\phi(x,p)}^{(\star)}(\cdot) \equiv \{\{\phi(x,p),\cdot\}\}$ is the adjoint action of the $\star$-algebra. Assuming $\phi(x,p)$ varies slowly in space, this becomes

$$(e^{-i\phi}h_0 e^{i\phi})(x,p) \rightarrow \sum_{n=0}^{\infty} \frac{1}{n!}\Big(\text{ad}_{\phi(x,p)}^{(p)}\Big)^n h_0(p) \tag{66}$$

where $\text{ad}_{\phi(x,p)}^{(p)}(\cdot) \equiv \{\phi(x,p),\cdot\}_{\text{p.b.}}$. Following the notation $U = e^{-\phi}$ used in ref. [1], we get

$$(e^{-i\phi}H_0 e^{i\phi})(x,p) \rightarrow U^{-1}H_0 U. \tag{67}$$

where the right-hand side is to be interpreted as the adjoint action of the lie group that represents the canonical transformations. The same argument follows for the term with the time-derivative, so

$$(e^{-i\phi}(i\partial_t - H_0)e^{i\phi})(x,p) \rightarrow U^{-1}(\partial_t - H_0)U. \tag{68}$$

using the notation in Ref. [1], that the integral over the phase-space can be expressed as an inner-product and using Eq. (61), the action becomes

$$S[\phi] = \text{tr}\big[f_0 e^{-i\phi}(i\partial_t - H_0)e^{i\phi}\big] \rightarrow \int dt \left\langle f_0, U^{-1}(\partial_t - H_0)U \right\rangle \tag{69}$$

Note, this is not an equality since we assumed that $\phi(x,p)$ varies slowly in space. This assumption is equivalent to the fact that in the momentum representation, the field $\phi(k,k',t)$ is non-negligible only when $|k-k'| \ll k_f$. When calculating the correlation functions, it can also be understood as the small momentum limit, $|q_i| \rightarrow 0$.

# B Comparison of the correlation functions obtained from fermion loop diagrams and the bosonized theory

Here, we show that the three-point and four-point density correlation functions obtained from our bosonized theory matches the results from fermion loop calculations. First, we review the calculation of a fermion loop with $n$ external boson legs.

We begin with the $n$-point density correlation function defined as

$$C_{(n)}(q_1, \omega_1; \cdots; q_n, \omega_n) \equiv \langle \mathbb{T} \rho(q_1, \omega_1) \cdots \rho(q_n, \omega_2) \rangle_c. \tag{70}$$

The density operator is defined as $\rho(q, \omega) = \int_{k,\omega'} \psi^\dagger(k+q, \omega'+\omega) \psi(k, \omega')$. $C_{(n)}$ can be expressed as the sum over $(n-1)!$ unique fermion loops. For simplicity, we only consider the fermion loop that describes the scattering process $k \to k+q_1 \to k+q_1+q_2 \to \cdots \to k+q_1+\cdots+q_n \to k$ and denote its contribution as $\mathscr{F}_{(n)}$.

For convenience, we define $p_i \equiv q_1 + \cdots + q_i$ and $\Omega_i \equiv \omega_1 + \cdots + \omega_i$. Then,

$$\mathscr{F}_{(n)} = -i^n \int_{k,\omega} \prod_{i=1}^n G(k+p_i, \omega+\Omega_i). \tag{71}$$

The Green's function is defined as $G(k, \omega) = -i \langle T\psi(k, \omega)\psi^\dagger(k, \omega) \rangle = (\omega - \varepsilon_k + i\,\mathrm{sgn}\varepsilon_k)^{-1}$. Integrating over $\omega$ we get

$$\mathscr{F}_{(n)} = -i^{n+1} \int_k \sum_{i=1}^n f_0(k+p_i) \prod_{\substack{j=1 \\ j \neq i}}^n \frac{1}{\Omega_j - \Omega_i - \varepsilon(k+p_j) + \varepsilon(k+p_i)}. \tag{72}$$

## B.1 Three-point correlation function

Now, let us compare the $n = 3$ fermion loop result with the three-point correlation function obtained in Section 3.3. For convenience, we reproduce the result given by Eq. (42) below.

$$C_{(3)}(1, 2, 3) = -\int_k \Bigg( \frac{F^{(3)}(k+q_1, k+q_1+q_2, k)}{(\omega_3 - \varepsilon_k + \varepsilon_{k+q_1+q_2})(\omega_2 - \varepsilon_{k+q_1+q_2} + \varepsilon_{k+q_1})} $$
$$+ \frac{F^{(3)}(k+q_2, k+q_2+q_3, k)}{(\omega_1 - \varepsilon_k + \varepsilon_{k+q_2+q_3})(\omega_3 - \varepsilon_{k+q_2+q_3} + \varepsilon_{k+q_2})} $$
$$+ \frac{F^{(3)}(k+q_3, k+q_1+q_3, k)}{(\omega_2 - \varepsilon_k + \varepsilon_{k+q_1+q_3})(\omega_1 - \varepsilon_{k+q_1+q_3} + \varepsilon_{k+q_3})} + (2 \leftrightarrow 3) \Bigg) \tag{73}$$

We claim that the first three terms, which we label $\mathscr{B}_{(3)}$, corresponds to the fermion loop $\mathscr{F}_{(3)}$. We prove this by explicitly showing $\mathscr{B}_{(3)} = \mathscr{F}_{(3)}$.

First, let us express $\mathscr{B}_{(3)}$ using $p_i$ and $\Omega_i$. To do this, we need to shift the integration variables of the second and third terms by $k \to k+q_1$ and $k \to k+q_1+q_2$ respectively. Then, we use the fact that $q_i = p_i - p_{i-1}$ and $\omega_i = \Omega_i - \Omega_{i-1}$, where here the indices are defined mod 3. This gives us

$$\mathscr{B}_{(3)} = \int_k \Bigg( \frac{F^{(3)}(k+p_1, k+p_2, k+p_3)}{(\Omega_3 - \Omega_2 - \varepsilon_{k+p_3} + \varepsilon_{k+p_2})(\Omega_1 - \Omega_2 - \varepsilon_{k+p_1} + \varepsilon_{k+p_2})} $$
$$+ \frac{F^{(3)}(k+p_2, k+p_3, k+p_1)}{(\Omega_1 - \Omega_3 - \varepsilon_{k+p_1} + \varepsilon_{k+p_3})(\Omega_1 - \Omega_3 - \varepsilon_{k+p_1} + \varepsilon_{k+p_3})} \tag{74}$$
$$+ \frac{F^{(3)}(k+p_3, k+p_1, k+p_2)}{(\Omega_2 - \Omega_1 - \varepsilon_{k+p_2} + \varepsilon_{k+p_1})(\Omega_3 - \Omega_1 - \varepsilon_{k+p_3} + \varepsilon_{k+p_1})} $$

Next, let us expand the expression for the $F^{(3)}$ factors. In general,

$$F^{(3)}(k_a, k_b, k_c) = -f_0(k_b) + f_0(k_a)f_0(k_b) + f_0(k_c)f_0(k_a) + f_0(k_b)f_0(k_c) - 2f_0(k_a)f_0(k_b)f_0(k_c).$$
(75)

If we substitute this into Eq. (74), we get

$$\mathscr{B}_{(3)} = \int_k \Bigg[ -\sum_{i=1}^{3} f_0(k+p_i) \prod_{\substack{j=1 \\ j \neq i}}^{3} \frac{1}{\Omega_j - \Omega_i - \varepsilon_{k+p_j} + \varepsilon_{k+p_i}}$$
$$+ \Big( f_0(k+p_1)f_0(k+p_2) + f_0(k+p_2)f_0(k+p_3) + f_0(k+p_3)f_0(k+p_1)$$
$$- 2f_0(k+p_1)f_0(k+p_2)f_0(k+p_3) \Big) \underbrace{\sum_{i=1}^{3} \prod_{\substack{j=1 \\ j \neq i}}^{3} \frac{1}{\Omega_j - \Omega_i - \varepsilon_{k+p_j} + \varepsilon_{k+p_i}}}_{=0} \Bigg].$$
(76)

It is straightforward to verify that the last factor vanishes by explicitly writing it out. It is a sum of products of fractions, and if we rewrite it as a single fraction with a common denominator, the numerator will be zero. Therefore, we are left with only the first term in the square brackets. A quick comparison with Eq. (72) shows us that this is the expression for $\mathscr{F}_{(3)}$:

$$\mathscr{B}_{(3)} = \mathscr{F}_{(3)} = -\int_k \sum_{i=1}^{3} f_0(k+p_i) \prod_{\substack{j=1 \\ j \neq i}}^{3} \frac{1}{\Omega_j - \Omega_i - \varepsilon_{k+p_j} + \varepsilon_{k+p_i}}.$$
(77)

## B.2 Four-point correlation function

Let us now check that the four-point correlation function obtained in Section 3.4 matches the $n = 4$ fermion loop result. For convenience, we reproduce the result given by Eq. (53) below.

$$C_{(4)}(1,2,3,4) = \Bigg( \underbrace{-\frac{i}{2} \sum_{i=1}^{4} \int_k g_{i+3,i+2}(k)g_{i+2,i+1}(k)g_{i+1,i}(k)}_{\equiv \mathscr{B}_{(4),1}} + \underbrace{i \sum_{i=1}^{4} \int_k g_{i+1,i}(k)g_{i+2,i}(k)g_{i+3,i}(k)}_{\equiv \mathscr{B}_{(4),2}}$$
$$\underbrace{-i \sum_{i=1}^{2} \int_k g_{i+3,i+2}(k)g_{i+2,i}(k)g_{i,i+1}(k)}_{\equiv \mathscr{B}_{(4),3}} \underbrace{-i \sum_{i=1}^{2} \int_k g_{i+2,i+3}(k)g_{i+3,i+1}(k)g_{i+1,i}(k)}_{\equiv \mathscr{B}_{(4),4}} \Bigg)$$
$$+ (\text{permutations of } \{(\omega_1, q_1), (\omega_2, q_2), (\omega_3, q_3), (\omega_4, q_4)\})$$
(78)

where

$$g_{i,j}(k) = \frac{f_0(k+p_i) - f_0(k+p_j)}{\Omega_i - \Omega_j + \varepsilon_{k+p_i} - \varepsilon_{k+p_j}}.$$
(79)

Since we are only interested in terms that we want to compare with $\mathscr{F}_{(4)}$, we can ignore the permuted terms and only consider the contributions from the first four terms of Eq. (78) which we label $\mathscr{B}_{(4)}$. The four terms have been separately labelled $\mathscr{B}_{(4),i}$ for $i = 1, 2, 3, 4$. Below, we show that $\mathscr{B}_{(4)} = \sum_{i=1}^{4} \mathscr{B}_{(4),i} = \mathscr{F}_{(4)}$.

Let us first look at $\mathcal{B}_{(4),2}$. The denominator of its integrands can be rewritten as

$$
\left( \text{Denominator of } g_{i+1,i}(k)g_{i+2,i}(k)g_{i+3,i}(k) \right) \tag{80}
$$

$$
= (\Omega_{i+1} - \Omega_i + \varepsilon_{k+p_{i+1}} - \varepsilon_{k+p_i})(\Omega_{i+2} - \Omega_i + \varepsilon_{k+p_{i+2}} - \varepsilon_{k+p_i})(\Omega_{i+3} - \Omega_i + \varepsilon_{k+p_{i+3}} - \varepsilon_{k+p_i})
$$

$$
= \prod_{\substack{j=1 \\ j \neq i}}^{4} \left( \Omega_j - \Omega_i + \varepsilon_{k+p_j} - \varepsilon_{k+p_i} \right). \tag{81}
$$

This is the denominator in $\mathcal{F}_{(4)}$. Let us now consider the numerator of the integrands in $\mathcal{B}_{(4),2}$.

$$
\left( \text{Numerator of } g_{i+1,i}(k)g_{i+2,i}(k)g_{i+3,i}(k) \right)
$$

$$
= (f_0(k+p_{i+1}) - f_0(k+p_i))(f_0(k+p_{i+2}) - f_0(k+p_i))(f_0(k+p_{i+3}) - f_0(k+p_i))
$$

$$
= -f_0(k+p_i) + f_0(k+p_i) \Big[ f_0(k+p_{i+1}) + f_0(k+p_{i+2}) + f_0(k+p_{i+3})
$$

$$
- f_0(k+p_{i+1})f_0(k+p_{i+2}) - f_0(k+p_{i+1})f_0(k+p_{i+3}) - f_0(k+p_{i+2})f_0(k+p_{i+3}) \Big] \tag{82}
$$

If we separate the first term of the above expansion from the rest, $\mathcal{B}_{(4),2}$ becomes

$$
\mathcal{B}_{(4),2} = i \sum_{i=1}^{4} \int_k (-f_0(k+p_i) + (\text{other terms})) \prod_{\substack{j=1 \\ j \neq i}}^{4} \frac{1}{\Omega_j - \Omega_i + \varepsilon_{k+p_j} - \varepsilon_{k+p_i}} \tag{83}
$$

$$
= \mathcal{F}_{(4)} + \underbrace{i \sum_{i=1}^{4} \int_k (\text{other terms}) \prod_{\substack{j=1 \\ j \neq i}}^{4} \frac{1}{\Omega_j - \Omega_i + \varepsilon_{k+p_j} - \varepsilon_{k+p_i}}}_{\equiv \mathcal{B}_{(4),2,R}}. \tag{84}
$$

Therefore, the first term of the expansion given in Eq. (82) gives us $\mathcal{F}_{(4)}$. The remaining terms in $\mathcal{B}_{(4),2}$ are collectively labelled $\mathcal{B}_{(4),2,R}$. Although tedious, it can then be shown by taking the integrands of $\mathcal{B}_{(4),2,R}, \mathcal{B}_{(4),1}, \mathcal{B}_{(4),3}, \mathcal{B}_{(4),4}$ and rewriting them as a single fraction with a common denominator that

$$
\mathcal{B}_{(4),2,R} + \mathcal{B}_{(4),1} + \mathcal{B}_{(4),3} + \mathcal{B}_{(4),4} = 0. \tag{85}
$$

Therefore,

$$
\mathcal{B}_{(4)} = \mathcal{B}_{(4),1} + \mathcal{B}_{(4),2} + \mathcal{B}_{(4),3} + \mathcal{B}_{(4),4} = \mathcal{F}_{(4)}. \tag{86}
$$

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
