# Peer review of "An exact method for bosonizing the Fermi surface in arbitrary dimensions"

_SciPost Physics_

## Round 1 · Referee Report · Anonymous · 2023-11-20

Report

In the manuscript "An exact method for bosonizing the Fermi surface in arbitrary dimensions", the authors present an exact approach for bosonization in higher dimensions. They offer an new derivation for the bosonization, different from the previous work by Delacretaz et al. This work is clear and solid. I really enjoy reading it. So I would like to recommend its publication in SciPost.

Below I list a few of my questions.

1. I think that the $H_0$ in Eq.2 is a second quantized Hamiltonian containing $\psi^{\dagger} \psi$ while the $H_0$ in Eq.6 and thereafter doesn't contain these operators.

2. In Eq.11, can I interpret "$\alpha$" physically as the number of patches (or the number of points on the Fermi surface)? Intuitively, $F_0$ is the total free energy so the action considered in Eq.11 is the averaged action per patch.

3. In the Eq.28, the 2n-th order of the action looks like describing n pairs of particle-hole excitation. Since it is an infinite sum, can we know which term is more relevant than others in the low energy? Or are they all equally important?

4. Is it possible to do RG on this action with, say, quartic interaction in Eq.25?

5. It seems that within your approach it is still hard to obtain any info about single fermion, e.g. propagator, anomalous dimension if there is any, if we start with the bosonic action. Is this correct? Does your method have any advantage towards this question?

6. It seems that the theory by Delacretaz et al corresponds to the low energy limit of yours, can you elaborate on what type of higher energy mode is absent in their theory but is included in yours?

  • validity: high
  • significance: good
  • originality: high
  • clarity: high
  • formatting: -
  • grammar: -

Author:  Takamori Park  on 2024-01-25  [id 4280]

(in reply to Report 2 on 2023-11-20)

We thank the referee for their valuable feedback and recommendation for publication. Below we list our responses to the suggestions and questions made by the referee:

  1. This is correct. We used second quantization in Eq. 2 to express $H_0$, but after that we switched to first quantization. An explanation that emphasizes this point was added to the manuscript.
  2. We believe that taking $\alpha$ does not describe a patch averaging process. In fact, when we take $\alpha\rightarrow0$, we get an exact result that contains all excitations which is in contrast to an average over patches.
  3. When calculating a $n$-th order correlation function, we only need to consider up to the n-th term in the action $S_{(n)}$ because higher-order terms generate diagrams with loops. In addition, the terms in the action that are relevant to calculating the n-th order correlation function are all equally important. For example, in the fourth order correlation function calculation, the contribution from $S_{(4)}$ (Eq. 52/Fig. 4e) is identical (up to an overall factor) to a term that doesn’t involve any higher-order terms from the action (Eq. 48/Fig. 4a). (Equation numbers correspond to the numbers in the resubmitted manuscript.)
  4. In principle, it is possible, but we have not tried it yet. This is something interesting that we can consider in the future.
  5. The equal-time Green’s function is given by $\langle f\rangle=\langle e^{i\phi}f_0e^{-i\phi}\rangle$. Naively, we can evaluate the expectation value of the exponential functions as the exponential of the propagator of $\phi$. This is how they are evaluated in 1D bosonization and how the anomalous dimension is obtained. However, this is not rigorous in our formalism because the action has higher order terms beyond the quadratic term and because of the restriction to tree diagrams. Therefore, indeed it is still difficult to calculate the anomalous dimension in d>1. We have included a discussion on this in the discussion section of the manuscript.
  6. Since we are considering non-interacting fermions in $d>1$, generally there are no sharp collective excitations and higher energy modes. However, it may be interesting to consider interactions to see what emergent excitations we can find.

---

## Round 1 · Referee Report · Anonymous · 2023-11-20

Report

In the manuscript “An exact method for bosonizing the Fermi surface in arbitrary dimensions”, the authors present a rigorous approach for expressing a fermionic theory with a Fermi surface as a constrained functional integral over a bosonic field. The authors established equivalence between the long-wavelength limit of their derived bosonized action and the action proposed by Delacretaz et al. The authors introduce a new perspective on the problem, yielding novel and intriguing findings. I have a few minor suggestions and questions aimed at strengthening its content.

1. The authors evaluated three- and four-point density correlation functions using proposed Feynman diagram rules, and the results from their bosonized theory precisely matched fermion loop diagram results. The exact matching of four-point density correlation functions is not sufficiently explained in the manuscript.
2. Could the authors provide additional commentary on how their approach may lead to corrections in the three- and four-point density correlation functions?
3. There are some minor typographical and grammatical errors in the manuscript that require attention.

In conclusion, I find the manuscript to be a valuable contribution to the field, and I recommend it for publication.

  • validity: -
  • significance: -
  • originality: -
  • clarity: -
  • formatting: -
  • grammar: -

Author:  Takamori Park  on 2024-01-25  [id 4281]

(in reply to Report 1 on 2023-11-20)
Category:
answer to question

We thank the referee for their valuable feedback and recommendation for publication. Below we list our responses to the suggestions and questions made by the referee:

  1. We added a new section to the appendix (Appendix B) that explicitly shows how to match the three- and four-point density correlation functions calculated in our formalism to the corresponding fermion loop results.
  2. The formalism we derived is exact and therefore, in principle, can be used to extract corrections to the asymptotic behavior in the long-wavelength limit. However, in practice, it is challenging.
  3. We have corrected typographical and grammatical errors made in the original manuscript.

---

## Round 1 · Referee Report · Anonymous · 2023-12-1

Report

The authors proposed a new method to obtain a bosonized action for fermions with Fermi surface. While their work mostly focused on non-interacting fermions, they also presented a promising avenue to include interaction effect. I found it interesting that their action can be reduced to a more phenomenological action considered earlier by Delacretaz et al in low energy limit. The paper is well written and informative. I recommend the publication of this article. I have a few suggestions/questions for improvement of the manuscript.

1) The authors showed how one can obtain the RPA result when only certain contributions are taken into account in the bosonized theory. In the fermionic theory, the RPA can be justified by taking large-N limit or high density limit. Is there a similar procedure or limit in the bosonized theory ?

2) It would be helpful to the readers if the authors explicitly show the results of three- and four-point density correlation functions computed in the fermionic theory and discuss the correspondence with the results from the bosonized theory. Then the discussion about the comparison between the bosonic theory and fermionic theory will be more explicit. At the moment, it is not easy to follow this discussion in the main text. One could add this information in the appendix.

  • validity: high
  • significance: good
  • originality: high
  • clarity: high
  • formatting: -
  • grammar: -

Author:  Takamori Park  on 2024-01-24  [id 4279]

(in reply to Report 3 on 2023-12-01)
Category:
answer to question

We thank the referee for their valuable feedback and recommendation for publication. Below we list our responses to the suggestions and questions made by the referee:

  1. There is no such new procedure or limit that we are currently aware of in our bosonized theory. The usual large-N limit can be used to justify the approximation we consider.
  2. We added a new section to the appendix (Appendix B) that explicitly shows how to match the three- and four-point density correlation functions calculated in our formalism to the corresponding fermion loop results.

---

## Editorial Decision

resubmitted